# GEES: Enabling Location Privacy-Preserving Energy Saving in Multi-Access Edge Computing

## Abstract

The global deployment of the 5G network has led to a substantial increase in the deployment of edge servers to host web applications, catering to the growing demand for low service latency by edge web users. Running edge servers 24/7 leads to enormous energy consumption and excessive carbon emissions. Energy-efficient edge resource provision is desired to achieve sustainable development goals in the new multi-access edge computing (MEC) architecture. Recently, several approaches have been proposed to solve the demand response problem for energy saving in cloud computing and MEC. However, accurate location information of edge web users should be provided, which sacrifices edge web users' privacy. To protect edge web users' location privacy while saving energy in MEC, we systematically formulate this location privacy-preserving edge demand response (LEDR) problem. To solve the LEDR problem effectively and efficiently, we propose a system named GEES by incorporating differential geo-obfuscation to secure user privacy, while maximizing system utility and energy efficiency through inference with theoretical analysis. Extensive and comprehensive experiments are conducted based on a synthetic real-world dataset, and the results demonstrate that GEES outperforms representative approaches by 23.02%, 31.47%, and 17.29% on average in terms of energy efficiency, user privacy and system utility.

## Keywords

multi-access edge computing, location privacy, edge energy saving

## 1 Introduction

The proliferation of data-intensive applications and the growing demands for low-latency services have highlighted the significance of Multi-Access Edge Computing (MEC) and reshaped the computing paradigms [1]. By redistributing resources to the network edges, MEC liberates the computing capabilities of web users' devices and enables real-time web services to improve users' quality of experiences. The evolving landscape of MEC systems has spurred research efforts across diverse domains, including edge intelligence [2, 3], edge privacy and security [4, 5], and edge resource [6, 7], driven by the goal of delivering low-latency web services [8].

Demand response in cloud computing has received extensive research attention [9–11], driven by predictions that global electricity consumption by data centers will reach 8% by 2030 [12]. As an extension of cloud computing, MEC involves dense edge server deployments near base stations, e.g., the density could reach as high as 50 per $km^2$ in real-world 5G deployments [13]. This results in significant *energy issue* with cost escalations for energy consumption, posing new challenges to environmental sustainability [14]. Recent efforts [15, 16] have begun addressing the edge demand response challenge by adopting strategies similar to those used in data centers — workload shifting and consolidation on selected edge servers, along with the shutdown of unused servers to conserve

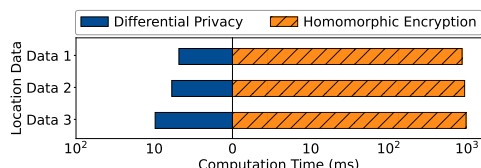

**Figure 1: Location privacy protection latency tests**

energy. To optimize the energy consumption of edge servers and satisfy users' quality of experience, users' actual locations are required by existing approaches. However, this greatly compromises users' location privacy. From the perspective of users, sacrificing their privacy to reduce the energy consumption of edge servers is a lack of motivation. This is undoubtedly disheartening, as it could lead to a reduction in users' willingness to access edge web services due to privacy concerns, ultimately eroding the benefits of the MEC architecture. Thus, it is urgent to solve this *privacy issue*.

To protect users' location privacy, cloaking-based methods[17] and homomorphic encryption [18] are two widely-used approaches. However, applying these techniques in edge demand response scenarios faces significant practical challenges. Cloaking-based methods aim to protect privacy by injecting obfuscated unreal data. However, severe distortion in cloaking unreal data leads to poor service performance [17]. Homomorphic encryption, which facilitates encrypted computations without exposing raw location data, introduces substantial computational complexity and serious overhead. This violates the key objective of MEC to pursue the low latency, causing performance degradation. Moreover, homomorphic encryption and cloaking-based methods are also sensitive to prior knowledge, rendering them vulnerable to inferences, such as Bayesian Inference Attacks (BIA) [19]. Practically, edge servers might attempt to infer users' locations by leveraging prior knowledge of users, with the intention of enhancing services, if they are aware that the uploaded locations are intentionally obfuscated.

Differential privacy [20] is another widely-used privacy-preserving approach in web services [21, 22], mobile crowdsensing (MCS) [23] and location-based services (LBS) [24] by introducing noise to location data to enable confident data sharing and build trust in location-based applications. Compared to homomorphic encryption, differential privacy is an extremely efficient method while ensuring absolute privacy for users. To demonstrate this, we measured the computation time of differential privacy and holomorphic encryption for a bigint transformed string containing the user's location information in August 2023. In the experiments, we apply homomorphic encryption[1] and differential privacy by adding noise drawn from Laplace distribution to the original data. The overall computation time reported in Figure 1 depicts the superior advantage of differential privacy safeguarding latency-sensitive services, e.g., the time taken by homomorphic encryption is > 100× the time taken by differential privacy. Compared to cloaking-based methods,

---

[1]https://github.com/data61/python-paillier

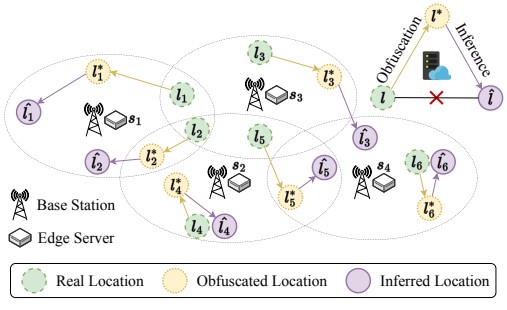

Figure 2: MEC system example

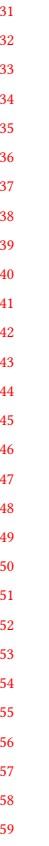

differential privacy offers a more rigorous and adaptable privacy protection framework, which demonstrates stronger robustness anti prior knowledge and a better trade-off between privacy and data availability [25]. Those advantages make differential privacy more suitable for addressing privacy-related problems in MEC than homomorphic encryption and cloaking-based methods.

Relying solely on differential privacy is still very challenging to realize an effective edge demand response strategy to achieve low energy consumption and high user coverage (referred to as *system utility* hereafter) due to the injected data noise. Figure 2 demonstrates a simple MEC system example comprising 4 base stations, 4 edge servers $\{s_1, ..., s_4\}$ and 6 users located at $\{l_1, ..., l_6\}$. Assuming that each edge server has enough resources, e.g., CPU, GPU, Memory, etc., to host nearby users. According to users' actual locations, all six users can be allocated to edge servers $s_2$, $s_3$ and $s_4$. In this case, edge server $s_1$ can be powered off to save energy. To protect location privacy based on differential privacy, base stations store the obfuscation mechanism provided by their trustworthy control server in advance and send the mechanism to users. In this way, users' locations $\{l_1, ..., l_6\}$ are obscured to obfuscated locations $\{l_1^*, ..., l_6^*\}$. Now, all four edge servers need to be powered on to serve nearby users based on $\{l_1^*, ..., l_6^*\}$, increasing the energy consumption. The inference method might be adopted by edge servers, where inferred locations are denoted as $\{\hat{l}_1, ..., \hat{l}_6\}$. According to inferred locations, $s_3$ can be powered off, since all the users can be allocated to $s_1$, $s_2$ and $s_4$. Although energy consumption is reduced, user $u_3$ is not truly covered by any running edge servers, thus further decreasing the system utility. Hence, developing a system that empowers users to utilize edge web services without privacy compromises while maintaining satisfactory system utility and energy efficiency, becomes increasingly imperative and urgent.

To address this location privacy-preserving edge demand response (LEDR) problem, considering energy and privacy issues in MEC, we design a **g**eo-obfuscated **e**dge **e**nergy **s**aving (GEES) system to enable fast location privacy preservation with the maximum system utility and energy efficiency jointly. Contributions of this paper can be summarized as follows:

- To the best of our knowledge, this is the first attempt to leverage differential geo-obfuscation in the edge demand response problem, so as to secure users' location privacy, ensure system utility and improve energy efficiency.
- GEES deploys a novel approach named Deflected Distribution Positioning (DDP) to maximize system utility with privacy guarantees by solving the Optimal Transport (OT) problem between the obfuscated distribution and the real distribution.

- Once users' locations are obfuscated, GEES employs a Secure Greedy Response (SGR) algorithm to finalize the edge demand response strategies by powering on and off the edge servers to improve system utility and energy efficiency.
- Comprehensive experiments verify that GEES significantly outperforms benchmarks, including the state-of-the-art approach with various differential privacy approaches.

## 2 Preliminaries

### 2.1 Differential Privacy

Differential privacy is a rigorous privacy protection scheme, which was first used for location privacy-preserving in [24]. It obfuscates the location through probabilistic means to prevent adversaries from distinguishing the users' actual locations. In this way, even if an adversary observes the published location $l^*$ and possesses the obfuscation mechanism $\mathcal{P}$, it fails to find the actual location $l$.

DEFINITION 1 (DIFFERENTIAL PRIVACY). *Suppose the obfuscated area includes a set of locations* $\{l_1, ..., l_n\} \in \mathcal{L}$, *then a probabilistic geo-obfuscation function P satisfies $\epsilon$-differential privacy, iff*

$$\mathcal{P}(l^* \mid l_1) < e^{\epsilon \cdot dist(l_1, l_2)} \cdot \mathcal{P}(l^* \mid l_2), \quad \forall l_1, l_2, l^* \in \mathcal{L} \quad (1)$$

*where $\mathcal{P}(l^* \mid l_1)$ is the probability of obfuscating $l_1$ to $l^*$, $dist(l_1, l_2)$ is the Euclidean distance between $l_1$ and $l_2$, and $\epsilon$ is the privacy budget. The smaller $\epsilon$, the higher the privacy.*

Exponential mechanism is a prevalent mechanism to realize $\epsilon$-differential privacy [26]. Given the overall location set $\mathcal{L}$ and neighboring protected location set (NPLS) $\zeta$, it leverages a scoring function $\rho : \mathcal{L} \times \zeta \rightarrow \mathbb{R}$, which assigns a real-valued score to each in/output pairs, in order to ensure that outputs with better utility can receive higher scores.

DEFINITION 2 (SENSITIVITY [27]). *For any pair of neighboring locations $l_1$ and $l_2$ on NPLS $\zeta$ and $l \in \mathcal{L}$, the sensitivity of the scoring function $\rho$ is given by its maximum change: $\triangle\rho = \max_{l_1, l_2, l} \mid \rho(l_1, l) - \rho(l_2, l) \mid$.*

DEFINITION 3 (EXPONENTIAL MECHANISM [28]). *Given a scoring function $\rho$ on $\mathcal{L} \times \zeta$ and actual location $l \in \mathcal{L}$, the exponential mechanism selects and outputs an element $l^* \in \zeta$ on NPLS with probability proportional to $\exp(\frac{\epsilon \cdot \rho(l, l^*)}{2\triangle\rho})$.*

To preserve $\epsilon$-differential privacy, other mechanisms, such as the Laplace mechanism and Gaussian mechanism, achieve geo-obfuscation by adding proportional noise drawn independently and identically distributed from the respective distributions to the query output.

### 2.2 Optimal Transport

Optimal transport (OT) is a mathematical framework that was originally developed to transport a set of resources from one location to another efficiently [29]. It serves as a powerful means of comparing and aligning probability distributions, enabling precise quantification of dissimilarities. In recent years, it has garnered significant attention in the computing field [30]. Compared to metrics such as $X^2$ and Kullback-Leibler divergence, it has demonstrated notable performance enhancements.

**Wasserstein distance** $\mathcal{W}_p$, as a key definition of OT, quantifies the dissimilarity between two probability distributions. By considering both the values and spatial arrangement of the distributions, it enables precise measurement of distributional discrepancies. This metric has been widely applied in probability-based applications, such as domain adaption [31], and data association [32].

DEFINITION 4 (*p*-WASSERSTEIN DISTANCE). *The p-Wasserstein distance between two probability measures $\mu$ and $\nu$ on a metric space $(X, d)$ is defined as $\mathcal{W}_p(\mu, \nu) = \left( \inf_{\gamma \in \Pi(\mu, \nu)} \int |x - y|^p \, d\gamma(x, y) \right)^{\frac{1}{p}}$, where $\mu$ and $\nu$ represent probability measures, $\Pi(\mu, \nu)$ is the set of all joint distributions with marginals $\mu$ and $\nu$, and $\gamma(x, y)$ represents the probability of the joint occurrence of $x$ and $y$. The parameter $p$ controls the emphasis on different characteristics of the distributions.*

Specifically, $\mathcal{W}_2$ considers individual contributions in 2-norm space, which is favored for its precision and computational efficiency in practical OT problems [33].

## 3  Problem Formulation

In this section, we first introduce the privacy games in MEC. Next, we present the system model and formally formulate the location privacy-preserving edge demand response (LEDR) problem.

### 3.1  Edge Privacy Games

Users and edge servers always have divergent interests when it comes to privacy issues in MEC. Users usually prefer not to disclose their actual location precisely when accessing edge services, while edge servers desire their actual locations to deliver accurate service and reduce energy consumption. Therefore, users and edge servers are often engaged in privacy games. Under edge privacy games, there are four main roles:

**Control server.** Hosted by the network provider with dedicated communication protocols and strict security measures for managing base stations, the control servers are regarded as inherently trustworthy entities [34, 35]. In this study, the control server generates the geo-obfuscation mechanism $\mathcal{P}$ and sends it to base stations for dispatching in advance (**Step #1** in Section 4.1).

**Base stations.** Base stations store $\mathcal{P}$ locally from the control server and play the role of senders. Once a user enters the coverage of a base station, the base station directly distributes $\mathcal{P}$ to her (**Step #2** in Section 4.1). We assume that base stations are also trustworthy, the same as [34, 35].

**Users.** Users receive $\mathcal{P}$ from base stations to obfuscate their actual locations locally to protect their privacy and upload obfuscated locations to edge servers (**Step #3** in Section 4.1).

**Edge servers.** Edge servers collaboratively serve nearby users. However, edge servers are semi-trusted, as they are interested in users' actual locations to achieve low-energy edge demand response and apply inference methods to predict such locations (Section 4.2).

Figure 3 illustrates a general architecture of edge privacy games:

**Geo-obfuscation in user-side.** From a user's perspective, her location $l_k$ needs to be obfuscated to obtain the new location $l_k^*$ through geo-obfuscation mechanism $\mathcal{P}$. The input of $\mathcal{P}$ is the actual location $l_k \in \mathcal{L}$, where $\mathcal{L}$ is the set of all possible values of the user's location, i.e., the locations that user could visit while releasing obfuscated location $l_k^* \in \mathcal{L}$ is sampled according to the following

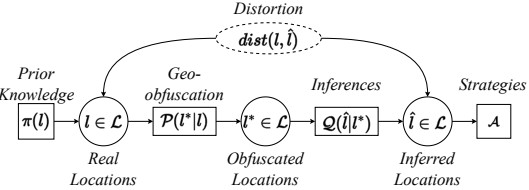

**Figure 3: Private game architecture**

probability distribution:

$$\mathcal{P}(l_k^* \mid l_k) = Pr\{l_k^* \in \mathcal{L} \mid l_k \in \mathcal{L}\} \geq 0 \ \ \& \ \ \sum_{l_k^* \in \mathcal{L}} \mathcal{P}(l_k^* \mid l_k) = 1$$

In this way, users' location privacy can be successfully protected by involving noise into the location information through the geo-obfuscation mechanism $\mathcal{P}$.

**Inference in server-side.** In fact, while the user is protecting her privacy, the edge server also makes efforts to reveal the user's actual location through inferences to deliver accurate service and reduce energy consumption on the edge servers. Let $\pi$ denote the prior leakage of the users' location distribution over $l$ as the prior knowledge: $\pi(l_k) = Pr\{l_k \in \mathcal{L}\}$. Based on $\pi(l_k)$, the edge server performs inferences on the obfuscated location $l_k^*$ from the user through an inference mechanism $Q$, aiming to find the users' actual location by observing the outcomes of the protection mechanism. Given any released $l_k^*$ and any prior knowledge of users, mechanism $Q$ generates estimated location $\hat{l}_k \in \mathcal{L}$ on edge servers according to the probability distribution:

$$Q(\hat{l}_k \mid l_k^*) = Pr\{\hat{l}_k \in \mathcal{L} \mid l_k^* \in \mathcal{L}\} \geq 0 \qquad (2)$$

### 3.2  System Model

Given $M$ physical edge servers $\mathcal{S} = \{s_1, ..., s_j, ..., s_M\}$ possessing $q$-dimensional computing resources $\mathcal{R} = \{r_1, ..., r_M\}$ ($r_j = \{r_j^1, ..., r_j^q\}$), $N$ users $\mathcal{U} = \{u_1, ..., u_k, ..., u_N\}$ with $h$-dimensional resources demands $d = \{d_1, ..., d_N\}(d_k = \{d_k^1, ..., d_k^h\})$, the objective of the location privacy-preserving edge demand response (LEDR) problem is to maximize the overall system performance, in terms of system utility $\mathcal{F}_u$, user privacy $\mathcal{F}_p$, and energy efficiency $\mathcal{F}_e$.

**System Utility.** A primary goal of MEC systems is to serve the maximum number of nearby users. Here, we define the system utility by the coverage rate of all the users in an MEC system, denoted by $F_c$. To calculate the system utility, we first introduce the effective user set in the LEDR problem:

DEFINITION 5 (EFFECTIVE USER SET). *Users with accessibility to edge web services whose real and inferred locations are within the range of a running edge server) are defined as set $\Delta X = \{u_k \mid dist(\hat{l}_k, s_j) \leq c_j \wedge dist(l_k, s_j) \leq c_j\}$, where $s_j$ is the edge server serving user $u_k$.*

According to the inferred location $\hat{l}$, the adjacent servers covering the inferred location $\hat{l}_k$ of user $u_k$, denoted by $\mathcal{J}(\hat{l}_k) = \{s_j \mid dist(\hat{l}_k, s_j) \leq c_j\}$. In this way, we define user $u_k$'s allocation decision as $a_k$, where $a_k \in \{0\} \cup \{j \mid s_j \in \mathcal{J}(l_k)\}$. Here, $a_k = j$ indicates that user $u_k$ is served by edge server $s_j$, while $a_k = 0$ means that $k$ is not allocated to any edge server. The user allocation strategy is denoted by $\mathcal{A} = \{a_1, ..., a_N\}$.

Let $\mathcal{N}_j(\mathcal{A})$ denote the number of users served by edge server $s_j$ according to $\mathcal{A}$, The total number of users covered by strategy

(a) Effective User      (b) Privacy Area

**Figure 4: Definitions of effective user and privacy area**

$\mathcal{A}$, denoted by $\mathcal{N}_{total}$ is calculated with $\mathcal{N}_{total} = \sum_{s_j \in \mathcal{S}} \mathcal{N}_j(\mathcal{A})$. Therefore, the system utility $F_c$, i.e., the coverage rate of all the users in an MEC system, is calculated with:

$$\mathcal{F}_c = \frac{\mathcal{N}_{total}}{N} \in [0, 1] \tag{3}$$

**User Privacy.** Following the analysis in [36], we first introduce $\mathcal{V}_k$, the inference error, i.e., the expectation of Euclidian distance between the $u_k$'s actual location $l_k$ and the inferred location $\hat{l}_k \in \zeta$:

$$\mathcal{V}_k = \sum_{\hat{l}_k \in \mathcal{L}} Q(\hat{l}_k \mid l_k^*) \cdot \|\hat{l}_k - l_k\|_2 \tag{4}$$

The inference error $\mathcal{V}_k$ can be considered how distorted the location can be restored during the inference process. Intuitively, we use the area proportion of inference error to the coverage radius $c_j$ of the allocated edge server $s_j$ to define the privacy rate $R$ to determine user privacy, as shown in Figure 4(b). In this way, a user $u_k$'s privacy $\mathcal{G}_k$ is calculated with:

$$\mathcal{G}_k = \| \frac{\mathcal{V}_k}{c_j} \|_2^2 \tag{5}$$

Therefore, the system privacy $\mathcal{F}_p$, calculated by the average privacy value of all the users, is:

$$\mathcal{F}_p = \sum_{u_k \in \Delta X} \frac{\mathcal{G}_k}{\mathcal{N}(a_k)} \in [0, 1] \tag{6}$$

**Energy Efficiency.** Edge demand response aims to reduce the energy consumption of edge servers while serving nearby users. However, the LEDR problem can be more complicated, as geo-obfuscation is introduced for privacy protection, and extra energy may be consumed as discussed in Section 1 with Figure 2. Therefore, how to maintain satisfying energy efficiency with privacy and utility guarantees is the critical challenge.

Similar to [6, 37], the energy consumption consists of three main components, including running cost $e_j^{rc}$, mode switching cost $e_j^{sc}$ and maintenance cost $e_j^{mc}$. Specifically, the running cost $e_j^{rc}$ can be calculated by $e_j^{rc} = \sum_{u_k \in \mathcal{N}_j(\mathcal{A})} p_k$, where $p_k$ is the energy cost for serving user $u_k$. The mode switching cost is calculated by $e_j^{sc} = \beta_j(1 - m_j)$, where $\beta_j$ is the start-up energy cost of activating server $s_j$, and $m_j \in \{0, 1\}$ represents current status of $s_j$, i.e., $m_j = 1$ is active and $m_j = 0$ is inactive. For the maintenance cost, $e_j^{mc} = \tau \cdot m_j$, where $\tau$ is the unit cost for maintaining one activated server. Then, the effective system energy consumption $E_c$ is denoted by:

$$E_c = \sum_{u_k \in \Delta X, s_j \in \mathcal{S}} e_j^{rc} + e_j^{sc} + e_j^{mc} \tag{7}$$

while the original energy consumption $E_o$ without edge demand response strategies is:

$$E_o = \sum_{u_k \in \mathcal{U}, s_j \in \mathcal{S}} e_j^{rc} + e_j^{sc} + e_j^{mc} \tag{8}$$

The system energy efficiency $F_e$ is defined by the ratio of effective energy consumption over the original energy consumption:

$$\mathcal{F}_e = \frac{E_c}{E_o} \in [0, 1] \tag{9}$$

## 3.3 Problem Formulation

Now, we formulate the Location privacy-preserving Edge Demand Response (LEDR) problem systematically by two phases: *differential geo-obfuscation* and *distortion-aware edge demand response*, jointly optimizing user privacy, system utility and energy consumption.

### 3.3.1 Probabilistic Differential Geo-obfuscation:

In the LEDR problem, edge servers may possess significant prior knowledge about the user geo-distribution within the region. For example, in an organized and vibrant area, users tend to be uniformly distributed [38]. Simultaneously, as introduced in Section 2.1, probabilistic methods could be employed to obfuscate users' locations. Such obfuscated locations uploaded from users to base stations for resource demands are often drawn from such an exponential distribution (details about selecting exponential distribution can be found in Section 4.1) under differential privacy [28]. Logically, the closer the obfuscated distribution aligns with the original distribution, the better the system's utility could be. Therefore, the first phase of the LEDR problem is to handle the gap between obfuscated distribution $\Phi$ with prior knowledge $\pi$ about users' real distribution while ensuring differential privacy guarantees. In this way, **Objective #1** of the LEDR problem is to minimize the Wasserstein-2 distance $\mathcal{W}_2$ between the two distributions:

$$\textbf{Objective \#1} \quad \min \ \mathcal{W}_2(\Phi, \pi) \tag{10}$$

$$s.t. \quad \mathcal{P}(l_k^* \mid l_1) < e^{\epsilon \cdot d(l_1, l_2)} \cdot \mathcal{P}(l_k^* \mid l_2), \quad \forall l_1, l_2, l_k^* \in \mathcal{L} \tag{11}$$

$$\sum_{l_k \in \mathcal{L}} \pi(l_k) \mathcal{P}(l_k^* \mid l_k) = \pi(l_k^*), \quad \forall l_k^* \in \mathcal{L} \tag{12}$$

As introduced in Sec. 2.1, constraint (11) ensures users' differential privacy with privacy budget $\epsilon$. Constraints (12) ensure that the geo-obfuscation will not change users' overall location distribution. Maintaining crucial statistics unchanged during obfuscation is a widely adopted approach in statistical disclosure control [23, 39]. It helps protect individual privacy while retaining general information in the obfuscated data, making it harder to infer specific individual details.

### 3.3.2 Distortion-aware Edge Demand Response:

The second phase of solving the LEDR problem is to allocate contained resources on edge servers to users under geo-distortion to maximize user privacy, system utility, and energy efficiency jointly. To generalize the LEDR problem, a set of parameters $\mathcal{B} = \{b_1, b_2, b_3\} (\sum_1^3 b_i = 1)$ are set based on the priority of system utility, user privacy, and energy efficiency in various MEC scenarios. This allows the model and the approach GEES (detailed in Section 4) be applied to various scenarios according to different needs. In this way, **Objective #2** of the LEDR problem is to maximize the overall system performance:

$$\textbf{Objective \#2} \quad \max \ b_1 \cdot \mathcal{F}_c + b_2 \cdot \mathcal{F}_p + b_3 \cdot \mathcal{F}_e \tag{13}$$

$$s.t. \quad \sum_{u_i \in \mathcal{U}} d_i^g \le r_j^g, \quad \forall g \le q, \ s_j \in \mathcal{S} \tag{14}$$

$$a_k \in \{0\} \cup \{j \mid dist(l_k, s_j) \le c_j\}, \quad \forall u_k \in \mathcal{U}, \ s_j \in \mathcal{S} \tag{15}$$

Resource constraint (14) guarantees that the allocated resources of each edge server cannot exceed its available resources and coverage constraint (15) confines that each user only can directly access edge servers covering this user. The impacts of priority of system utility, user privacy, and energy efficiency, i.e., $\mathcal{B} = \{b_1, b_2, b_3\}$, are detailed

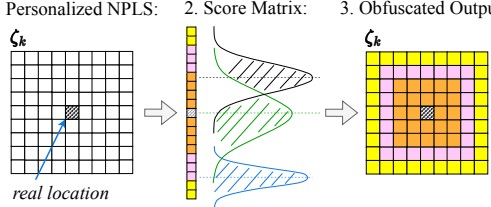

1. Personalized NPLS:    2. Score Matrix:    3. Obfuscated Outputs:

$\zeta_k$    $\zeta_k$

*real location*

**Figure 5: Probabilistic optimization process**

evaluated and discussed in Section 5.3. The problem hardness of LEDR is theoretically analyzed in Appendix B.

## 4 Geo-obfuscated Edge Energy Saving System

To tackle the LEDR problem systematically, including *geo-obfuscation probabilistic optimization* and *resource allocation under geo-distortion*, we design a geo-obfuscated edge energy saving (GEES) system to maximize user privacy, system utility and energy efficiency. Following the logic in Section 3.3, GEES first proposes a novel approach (Section 4.1) named deflected distribution position (DDP) to optimize geo-obfuscation probabilistic to achieve **Objective #1**, then designs a secure greedy response (SGR) approach (Section 4.2) to improve the overall system performance, i.e., **Objective #2**.

### 4.1 Geo-obfuscation Probabilistic Optimization

To achieve the geo-obfuscation function optimization, we propose an exponential-based probabilistic method, inspired by the Particle Swarm Optimization (PSO) [40]. Compared to other mechanisms in differential privacy, such as Laplace [24] and Gaussian [27] which directly add noise to location data, the exponential mechanism provides a more flexible form of randomness. By designing a scoring function to assess the utility of the data, it can offer more accurate privacy protection in various scenarios, allowing for a more precise trade-off between privacy and utility. Thus, the exponential mechanism is adopted in our design to perform geo-obfuscation. To evaluate the impacts of various differential privacy mechanisms, extensive experiments are conducted in Section 5, while the results can verify the advantages of the exponential mechanism.

As introduced in section 2.1, a scoring function $\rho$ should be defined to accommodate the LEDR scenario. The ideal situation is that the obfuscated location distribution uploaded to an edge server can closely resemble the actual location distribution of users while adhering to privacy constraints. Specifically, for a personalized NPLS, an appropriate scoring function would assign higher scores to the locations near the actual locations, aiming to keep the obfuscated locations as close as possible to the original ones. However, due to privacy and objective constraints, the mapping results often experience deflections.

To tackle the above-mentioned issue, a novel approach named Deflected Distribution Positioning (DDP) is designed in this section. DDP adopts the probability density values drawn from the Gaussian distribution $X \sim N(\mu, \sigma^2)$ as scores to simulate the corresponding outputs. By optimizing the $\mu$ and $\sigma^2$ under privacy constraints consistently, the position of Gaussian distribution is manipulated nimbly on the array to accommodate the requirements and generate corresponding scores wisely. Figure 5 provides an overview of the DDP algorithm 1, which consists of three steps.

- **Step #1**. Given $u_k$ actual location $l_k$, the NPLS around the actual location $\zeta_k$ is partitioned into a $10 \times 10$ location set on base stations, where each grid in this set corresponds to a $10m \times 10m$ unit. Then, personalized NPLSs $\zeta_k$ are transformed into $100 \times 1$ arrays. Each grid is assigned a score value drawn from the probability density function of a shiftable and adjustable Gaussian distribution $N(\mu, \sigma^2)$. In this step, a classical evolutionary algorithm of Particle Swarm Optimization (PSO) is employed to optimize the parameters of the distribution on the trustworthy base stations. As discussed in Section 3.1, this process is executed by the control server of base stations *in advance*. After the optimization, the control server sends mechanism $\mathcal{P}$ to base stations.
- **Step #2**. Once a base station receives $\mathcal{P}$, it stores $\mathcal{P}$ locally. After that, if a user enters its coverage, it automatically sends $\mathcal{P}$ to the user for geo-obfuscation.
- **Step #3**. In this step, users can apply $\mathcal{P}$ on their own devices to obfuscate their locations.

Finally, the obfuscated results are transmitted to edge servers to balance the privacy constraints and system utility. The pseudo-code of DDP is presented in Algorithm 1.

---

**Algorithm 1** Deflected Distribution Positioning

---

1: **initialization**
2:    set swarm of $I$ particles with bounded $\mu_i$ and $\sigma_i^2$ and $T$ iteration times
3:    set particle best positions and fitness: $\mathbf{p}_{global} = \mathbf{p}_{local} = (\mu_i, \sigma_i^2)$, $f_{global} = f_{local} = f(\mu_i, \sigma_i^2)$
4: **end initialization**

   as Trustworthy Control Server:
   ▷ **Step #1**: Preparation and Probabilistic Optimization
5: Partition the NPLSs and transform into $100 \times 1$ arrays
6: **for** each iteration $t = 1, ..., T$ **do**
7:    **for** each particle $i = 1, ..., I$ **do**
8:      Update particle velocity:
9:      $\Delta\mu_i = c_1^{\mu} \cdot \text{rand}() \cdot (\mu_{local} - \mu_i) + c_2^{\mu} \cdot \text{rand}() \cdot (\mu_{global} - \mu_i)$
10:      $\Delta\sigma_i^2 = c_1^{\sigma} \cdot \text{rand}() \cdot (\sigma_{local}^2 - \sigma_i^2) + c_2^{\sigma} \cdot \text{rand}() \cdot (\sigma_{global}^2 - \sigma_i^2)$
11:      Update particle position:
12:      $\mu_i = \mu_i + \Delta\mu_i$
13:      $\sigma_i^2 = \sigma_i^2 + \Delta\sigma_i^2$
14:      Evaluate fitness: $f_i = f(\mu_i, \sigma_i^2)$
15:      **if** $f_i > f_{local}$ **then**
16:        Update particle best: $\mathbf{p}_{local} = (\mu_i, \sigma_i^2)$, $f_{local} = f_i$
17:        **if** $f_i > f_{global}$ **then**
18:          Update global best: $\mathbf{p}_{global} = \mathbf{p}_{local}$, $f_{global} = f_i$
19:        **end if**
20:      **end if**
21:    **end for**
22: **end for**
23: Dispatch optimized exponential-based mechanism $\mathcal{P}$ to base stations

   as Base station:
   ▷ **Step #2**: Mechanism Storage and Dispatch
24: store $\mathcal{P}$ locally
25: send $\mathcal{P}$ to a user once the user enters its coverage

   as User $u_k$:
   ▷ **Step #3**: Geo-Obfuscation
26: $u_k$'s personalized geo-obfuscation with optimized $\mathcal{P}(l_k^*|l_k)$

---

## 4.2 Secure Greedy Response

Based on DDP proposed in Section 4.1, GEES now formulate a privacy-aware edge demand response strategy to solve the LEDR problem under geo-distortion. To address it, we propose a greedy-based mechanism named secure greedy response (SGR) for jointly maximizing user privacy, system utility and energy efficiency.

Given the NP-hardness of the problem, SGR employs a heuristic approach by consistently prioritizing user allocations with both the highest joint privacy and resource demands to the remaining available resources on nearby edge servers. Before introducing the details of SGR, key definitions are firstly detailed below:

**DEFINITION 6 (DOMINANT RESOURCE DEMANDS).** *Considering user $u_k$ with her q-dimensional resource demand $d_k = \{d_k^1, ...d_k^q\}$, the dominant resource demand $d_k^*$ of $u_k$ is the maximum proportion of required resources over the available resource of nearby edge servers, formulated as:*

$$d_k^* \triangleq \max_{r_j \geq d_k} \frac{d_k^g}{\max_{s_j \in \mathcal{J}(\hat{l}_k)} r_j^g} \quad (16)$$

**DEFINITION 7 (JOINT PRIVACY-ENERGY PRINCIPLE).** *Given edge server $s_j$, its coverage radius $c_j$, and the distance between the user's obfuscated location $l_k^*$ and the inferred location $\hat{l}_k$, the joint privacy-energy rate is defined as:*

$$y_j \triangleq \max_{r_j \geq d_k, s_j \in \mathcal{J}(\hat{l}_k)} \frac{(b_2 \cdot dist(\hat{l}_k, l_k^*) + 1) \cdot d_k}{b_3 \cdot r_j \cdot c_j} \quad (17)$$

*Consequently, the joint energy-privacy principle will therefore select the server $s_j$ with maximum $y_j$ for responding.*

The pseudo-code of SGR is presented in Algorithm 2. SGR starts with the initialization of unserved users $\mathcal{U} = \{u_1, ..., u_N\}$ and edge servers $\mathcal{S} = \{s_1, ..., s_M\}$ (Lines 1-3). SGR will first obtain users obfuscated locations through Algorithm 1 (Line 4). Next, SGR performs inferences and finds nearby edge servers based on the users' obfuscated locations $l^*$ and sorts their resource demands $\mathcal{D} = \{d_1, ..., d_N\}$ in decreasing order (Lines 5-6). After that, SGR iterates to formulate the edge demand response $\mathcal{A}$ by allocating users to available edge servers following the joint privacy-energy principle until no feasible allocation decision updates such as all users are allocated or no sufficient resources to serve any unserved users (Lines 7-21). In each iteration, the algorithm first initializes allocation decision $a_k$ as null (Line 9). Next, after checking the availability of nearby edge server based on $u_k$'s obfuscated location $l_k^*$, SGR iterates to allocate unserved $u_k$ to an available edge server $s_j (s_j \in \mathcal{J}(l_k^*))$ based on the joint privacy-energy principle and update the remaining resources of $s_j$ (Lines 10-14). After the iteration, the system information can be updated, including the available edge servers and remaining resources (Lines 15-18). Finally, $\mathcal{A}$ is returned for implementation as the strategy of this LEDR problem (Lines 22-23).

## 4.3 Theoretical Performance Analysis

DDP in Algorithm 1 consists of particle iteration and update processes. With $I$ particles, DDP iterates $T$ times for each user. During each iteration, every particle needs to update its mean and variance, calculate fitness, and perform comparisons. Therefore, the time complexity of Algorithm 1 is $O(I \cdot T \cdot O(1)) = O(I \cdot T)$

In Algorithm 2, SGR aims to formulate a series of allocation strategies with a maximum of $N$ iterations (Line 6). In each iteration,

---

**Algorithm 2** Secure Greedy Response

1: **initialization**
2: set with unserved users $\mathcal{U} = \{u_1, ..., u_N\}$ and available edge servers $\mathcal{S} = \{s_1, ..., s_M\}$
3: **end initialization**
4: obtain $\{l_1^*, ...l_k^*, ...l_N^*\} \in \mathcal{L}$ by Algorithm 1
5: obtain $\{\hat{l}_1, ...\hat{l}_k, ...\hat{l}_N\} \in \mathcal{L}$ through inferences and find all the nearby edge servers $\mathcal{J}(\hat{l}_k)$ of each estimated $\hat{l}_k$
6: sort users' demands in a decreasing order following dominant resource demands
7: **repeat**
8:   **for** $u_o \in \mathcal{U}$ **do**
9:     $a_k = (0, ..., 0)$ // $a_k^t = \{a_k^1, ..., a_k^h\}$
10:     **if** $\mathcal{J}(\hat{l}_k) \neq \emptyset$ and $u_k$ is unserved **then**
11:       find the most suitable edge server $s_j$ based on the joint privacy-energy principle.
12:       update $a_k$ with $a_k \leftarrow j$
13:       update $s_j$'s remaining resources with $r_j \leftarrow r_j - d_k$
14:     **end if**
15:     **if** $r_j < \min d_k$ **then**
16:       $\mathcal{S} \leftarrow \mathcal{S}/s_j$
17:     **end if**
18:     update the status of neighbor machines $\mathcal{J}(\hat{l}_k) \in \mathcal{S}$
19:     $k \leftarrow k + 1$
20:   **end for**
21: **until** no decision updates or no sufficient resources
22: **return** $\mathcal{A} = (a_1, ..., a_N)$
23: implement LEDR strategies $\mathcal{A}$

---

finding the optimal edge server for an individual user is $O(M)$ since there are a maximum of $M$ nearby edge servers for consideration (Line 9). Therefore, Algorithm 2 can formulate strategies within $O(N \cdot M)$ time in the worst-case scenario.

The theoretical analysis of user privacy achieved by GEES can be found in Appendix C.

## 5 Evaluation

In this section, experiments are conducted to evaluate the performance of GEES in LEDR scenarios. Both the dataset and experiment codes used have been published[2] for the validation and reproduction of experimental results.

## 5.1 Experiment Settings

**Experiment Data:** Here, we synthesize a new dataset named EDR based on the information from AWS wavelength[3], Alibaba Cloud[4], and a real-world dataset EUA[5], including edge server capacities, edge server coverages, edge server and user locations, server start-up and maintenance costs, etc.

**Impletations:** To comprehensively analyze the performance of our approach in various MEC scenarios, we conduct a series of experiments, i.e., Set #1, with variations in four parameters: 1) the number of users $N$; 2) the number of edge servers $M$; 3) the privacy budget $\epsilon$; and 4) weighted coefficients $\mathcal{B}$, respectively. Moreover,

---

[2]https://anonymous.4open.science/r/LEDR-8AB4
[3]https://aws.amazon.com/wavelength/features/
[4]https://github.com/alibaba/clusterdata
[5]https://github.com/swinedge/eua-dataset

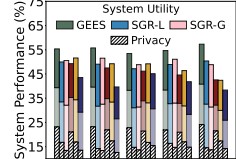 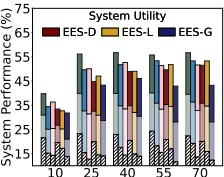 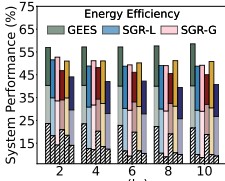 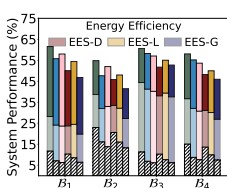

(a) System performance vs. N  (b) System performance vs. M  (c) System performance vs. $\epsilon$  (d) System performance vs. $\mathcal{B}$

**Figure 6: System effectiveness with geo-obfuscation**

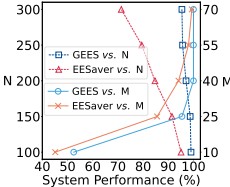

**Figure 7: System effectiveness without geo-obfuscation**

we evaluate the performance of GEES in terms of system utility and energy efficiency when no privacy protection mechanism is applied in Set #2 to demonstrate the universality of GEES. In the default configurations, users are uniformly distributed within the area, where each user's personalized NPLS is divided into a 10x10 grid, with each $10 \times 10$ unit size. Additionally, the weights coefficients are set as $\mathcal{B}_2 = (0.25, 0.5, 0.25)$ by default. Table 1 provides an overview of the experiment configurations. We repeat experiments 15 times each time a setting parameter changes and report the average results.

## 5.2 Benchmark Systems

To evaluate GEES, a state-of-the-art edge demand response approach called EESaver [16] is involved. EESaver aims to reduce energy consumption while maintaining efficient service provision in a greedy manner. However, EESaver focuses exclusively on tackling edge demand response issues, inadvertently neglecting user privacy concerns. For a fair comparison, EESaver is enhanced to **EESaver-D**, **EESaver-L** and **EESaver-G** with various privacy guarantees, i.e., DDP (in Section 4.1), Laplace [24] and Gaussian [27], respectively. To further investigate the impact of various privacy guarantees, we also combine our SGR algorithm (in Section 4.2) with Laplace and Gaussian, i.e., **SGR-L** and **SGR-G**, in the experiments for comparison. Details of Laplace and Gaussian differential privacy can be found in Appendix D.

## 5.3 System Effectiveness

Figure 6 demonstrates the performance of GEES in experiment Set #1.1, #1.2, #1.3, and #1.4 respectively, measured by the overall system score in Eq. (13). Different components of the scores in user privacy, system utility, and energy efficiency are presented individually to constitute the overall system performance. In general, the results show that *GEES consistently achieves the highest system performance in all cases*, and our SGR-based approaches, i.e., SGR-L and SGR-G, also outperform other approaches significantly across various experimental configurations and sets.

### Table 1: Parameter Settings

|  | Users $N$ | Edge Servers $M$ | Privacy $\epsilon$(ln) | Coefficients $\mathcal{B}$ |
|---|---|---|---|---|
| Set #1.1 | 100, 150, ..., 300 | 30 | 2 | $\mathcal{B}_2$ |
| Set #1.2 | 150 | 10, 25, ..., 70 | 2 | $\mathcal{B}_2$ |
| Set #1.3 | 150 | 30 | 2, 4, ..., 10 | $\mathcal{B}_2$ |
| Set #1.4 | 150 | 30 | 2 | $\mathcal{B}_1, \mathcal{B}_2, \mathcal{B}_3, \mathcal{B}_4$ |
| Set #2.1 | 100, ..., 300 | 30 | NA | $\mathcal{B}_5$ |
| Set #2.2 | 150 | 10, ..., 70 | NA | $\mathcal{B}_5$ |

where $\mathcal{B}_1 = (0.50, 0.25, 0.25)$, $\mathcal{B}_2 = (0.25, 0.50, 0.25)$, $\mathcal{B}_3 = (0.25, 0.25, 0.50)$, $\mathcal{B}_4 = (0.33, 0.33, 0.33)$, and $\mathcal{B}_5 = (0.50, 0.00, 0.50)$.

**Impact of system size**. In practical applications, ensuring scalability is of paramount importance. Thus, GEES must be able to scale with system size, including the number of users and the number of edge servers. With the increase in the number of users from 100 to 300 in Set #1.1, Figure 6(a) depicts that GEES significantly outperforms SGR-L, SGR-G, EESaver-D, EESaver-L, EESaver-G by 50.36%, 67.18%, 7.74%, 34.86%, 63.34% in terms of user privacy on average. This shows the superior performance of GEES in privacy preservation. In terms of system utility and energy efficiency, the performance of GEES reaches close to SGR-G, which achieves the highest system utility and energy efficiency. Notably, while the PSO-based EESaver (EESaver-P) approach ensures the second-highest privacy, its usability significantly lags behind GEES by large margins. With an increasing number of edge servers (Set #1.2), Figure 6(b) demonstrates that the overall system performance of all approaches increases from 10 to 25, and stabilizes when the number of edge servers exceeds 25. In addition, GEES continues to achieve the highest system performance, especially user privacy, with competitive system utility and energy efficiency.

**Impact of privacy budget**. Experimental results of Set #1.3 demonstrate the variations in the privacy budget $\epsilon$ from ln2 to ln10. Figure 6(c) depicts that user privacy achieved by all the approaches slightly decreases while the system utility and energy efficiency achieved by all the approaches increase, with the increase in the privacy budget $\epsilon$. Serving as a pivotal parameter in differential privacy, $\epsilon$ quantifies the intensity of privacy protection, where larger $\epsilon$ corresponds to weaker privacy levels. Consequently, as $\epsilon$ increases, the average user privacy decreases. Simultaneously, it grants the system greater flexibility in utilizing users' actual location, leading to better system utility and energy efficiency. Again, GEES achieves the highest user privacy and overall system performance. Comparatively, while the baselines can maintain relatively similar levels of privacy, their usability falls significantly short of GEES.

**Impact of weighted coefficients**. To evaluate the impacts of the weighted coefficients in Section 3.3.2, i.e., $\mathcal{B} = \{b_1, b_2, b_3\}$, Set #1.4 sets various combinations of those coefficients. As illustrated by Figure 6(d), under various configuration weights, GEES outperforms other comparison approaches again. Based on the analysis above, GEES has been proven to provide privacy protection for web users while maintaining satisfactory system utility and energy efficiency.

**Performance without privacy protection**. Now, we investigate the performance of GEES in scenarios without privacy protection in Sets #2.1 and #2.2. In this case, weighted coefficients are set to $b_1 = b_3 = 0.5$, as the privacy weight is $b_2 = 0$. The experimental results are shown in Figure 7. In general, GEES still outperforms EESaver with significant margins without privacy protection. In

Figure 7, as the number of users rises from 100 to 300, GEES consistently outperforms the EESaver in a stable way, with from 4.14% to 33.73% higher in the system performance. With the increase in the number of edge servers from 10 to 70, the system performance of both GEES and EESaver increases. Specifically, the system performance of GEES is 16.28% higher than that of EESaver when $M$ is 10. Apparently, even in LEDR scenarios without privacy consideration, GEES can still achieve higher performance, compared to the state-of-the-art approach.

### 5.4 System Overhead

As discussed in Section 1, achieving low latency is a primary goal of MEC. Consequently, the system overhead is an important metric to evaluate the performance of GEES, as LEDR strategies must be swiftly formulated to guarantee real-time service. Here, the system overhead is measured by the time taken to formulate LEDR strategies. Figure 8 shows the computation time taken by different approaches to formulate LEDR strategies. GEES and SGR-based approaches introduce an overhead of less than 0.2 seconds while EESaver-based approaches take around 1 second. Specifically, as the number of users rises from 100 to 300 in Figure 8(a), the computation time of GEES increases from 0.11s to 0.34s, and the SGR-based approaches maintain relatively lower computation times among these approaches. On the other hand, the EESaver-based approaches show a steep increasing trend from 0.64s to 1.59s. As the number of servers increased from 10 to 70 in Figure 8(b), the results exhibit a similar pattern, with the computation time changing from 0.07s to 0.34s for GEES and from 0.38s to 1.87s for EESaver-based approaches on average. Figure 8(c) demonstrates the computation time when the privacy budget increases. Moreover, the system overheads of all the approaches remain stable, while GEES consistently outperforms the EESaver-based approaches by 0.73s faster on average. Meanwhile, GEES demonstrates lower computational time across different sets of weight coefficients in Figure 8(d).

Based on the analysis above, among all the approaches, GEES is proven to solve the LEDR problem effectively and efficiently.

## 6 Related Work

### 6.1 Edge Demand Response

The proliferation of Multi-access Edge Computing (MEC) has opened a number of research topics, including edge-assisted federated learning [41], edge user allocation [15, 37] and edge data caching [42–44]. However, MEC faces new challenges in optimizing resource allocation in dynamic environments with the concerns of both the user privacy guarantee and the energy consumption. Recently, researchers start paying attention to energy saving in MEC. Chen et al. [6] propose an online auction mechanism via cloudlet control to optimize energy allocation and resource utilization. This approach solely takes into account the status of cloudlets, however, its scalability is limited by addressing resource constraints separately in real-world scenarios. Edgedr [37] surpasses the primary constraints above by granting the flexibility to control the energy consumption of each server individually. In pursuit of heightened performance and utility in MEC, Cui et al. [16] propose EESaver, a mechanism to reduce energy consumption while maintaining good system performance. However, EESaver falls short of addressing the growing

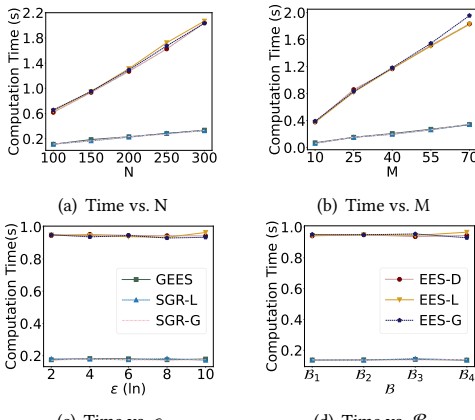

(a) Time vs. N       (b) Time vs. M

(c) Time vs. $\epsilon$       (d) Time vs. $\mathcal{B}$

**Figure 8: System Overhead**

privacy concerns of users in the MEC environment. In fact, there still exists a research gap to explore the privacy and energy issues in edge demand response.

### 6.2 Location Privacy

Location privacy preservation has emerged as a critical concern in order to adapt for location-sensitive applications, such as mobile crowdsensing (MCS) [45, 46] and local business service systems (LBSS) [47]. Cloaking [17, 48] and homomorphic encryption [49, 50] have been widely used in practice for protecting location privacy due to their feasibility. Regrettably, such methods might lead to significant location distortion and overhead in MEC with high latency, thereby compromising service quality. Meanwhile, the aforementioned methods are also sensitive to the adversary's prior knowledge. To mitigate this concern, more studies are involving differential privacy methods to protect location privacy. Wang et al. [23] introduce a new framework for task allocation in MCS, which integrates differential geo-obfuscation to safeguard participants' location privacy while optimizing worker travel distance. Yu et al. [28] investigate a personalized error-bounded dynamic differential location privacy mechanism to defend privacy leakage against Bayesian adversaries. In this paper, we leverage the advantages of differential privacy in the LEDR problem, aiming to ensure user privacy while maximizing system utility and energy efficiency.

## 7 Conclusion

This paper investigates the location privacy-preserving edge demand response (LEDR) problem. To tackle the LEDR problem systematically and theoretically, we propose a novel system, named GEES, to optimize the geo-obfuscation and edge demand response strategies collectively. GEES leverages differential geo-obfuscation to protect users' location privacy, simultaneously ensuring system utility and energy efficiency. GEES is verified via extensive experiments against representative approaches under various privacy mechanisms. Experimental results demonstrate that GEES outperforms other mechanisms with superior performance on user privacy, system utility, and energy efficiency. As part of our future work, we will delve into the potential vulnerabilities of GEES and explore corresponding defense mechanisms.

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

# A  Summary of Notations

**Table 2: Summary of Notations**

| Notation | Description |
|---|---|
| $a_k = j$ | decision to allocate $u_k$ to $s_j$ |
| $\mathcal{B} = \{b_1, b_2, b_3\}$ | pre-configured weighted coefficients |
| $c_j$ | coverage radius of $s_j$ |
| $\mathcal{D} = \{d_1, ..., d_N\}$ | set of user resource demands |
| $d_k = \{d_k^1, ..., d_k^h\}$ | $u_k$' h-dimensional resource demand |
| $\mathcal{G}_k$ | $k$'s privacy area proportion |
| $\mathcal{L}$ | actual location distribution sets |
| $l_k$ | $u_k$'s actual location |
| $l_k^*$ | $u_k$'s obfuscated location |
| $\hat{l}_k$ | inferred location of $u_k$ |
| $M$ | # of edge servers |
| $m_j$ | activation status of $s_j$ |
| $N$ | # of users |
| $\mathcal{N}_j(a_k)$ | # of users allocated to $s_j$ |
| $\mathcal{P}(l^* \mid l)$ | geo-obfuscation mechanism assigning $l$ to $l^*$ |
| $Q(\hat{l} \mid l^*)$ | inference mechanism given $l^*$ to infer $\hat{l}$ |
| $\mathcal{R} = \{r_1, ..., r_M\}$ | set of edge server resources |
| $r_j = \{r_j^1, ..., r_j^q\}$ | $s_j$'s q-dimensional resources |
| $\mathcal{S} = \{s_1, ..., s_M\}$ | set of edge servers |
| $s_j$ | edge server $j$ |
| $\mathcal{U} = \{u_1, ..., u_N\}$ | set of users |
| $u_k$ | user $k$ |
| $\mathcal{V}_k$ | $u_k$'s inference error |
| $\Delta X$ | effective user set |
| $\zeta$ | neighboring protected location set of users. |
| $\beta_j$ | start-up energy cost of activating $s_j$ |
| $\tau$ | cost of maintaining an activated server |
| $\epsilon$ | $\epsilon$-differential privacy budget |
| $\pi_k$ | prior knowledge of $u_k$'s overall location distribution |

# B  NP-hardness of LEDR

PROOF. Here, we systematically analyze the hardness of the LEDR problem. Without loss of generality, we decompose our LEDR problem into two sub-problems as discussed in 3.3, namely probabilistic differential geo-obfuscation (PDG) and distortion-aware demand response allocation (DDR). Afterward, we introduce the NP-hard Bin Packing (BP) problem, which aims to pack items into the minimum number of bins while respecting their capacities.

For the PDG sub-problem, let the sizes of the items be denoted as $\omega_i$ and the capacities of the bins by $\tau$. Given an instance of the classic BP problem, we construct two distributions $P$ and $Q$ as follows:

$$P = \{\omega_1, \omega_2, \ldots, \omega_n\}, \quad Q = \{\tau, \tau, \ldots, \tau\},$$

where $P$ represents the sizes of items and $Q$ represents the capacities of bins. The Wasserstein distance $\mathcal{W}(P, Q)$ between these distributions represents the minimum cost of redistributing the items (sizes in $P$) to satisfy the capacities (sizes in $Q$). Therefore, the above optimal transport problem remains NP-hard following the reduction.

Meanwhile, it is assumed that DDR problem is also a type of classic BP problem, involving $m$ bins $\mathcal{S} = \{s_1, ..., s_m\}$ and $n$ items $\mathcal{U} = \{u_1, ..., u_n\}$ with the size of $d_i$, where each bin are endowed with capacities $r_j$. The conventional BP problem is concerned with

arranging items into minimal bins while adhering to the constraint: $\sum_{u_i \in \mathcal{U}} d_i^h \leq r_j^h, \forall h \leq q, s_j \in \mathcal{S}$. In this scenario, edge servers and users are projected to bins and items, respectively. Therefore, it can be transformed into fill edge servers' available resources by users' resource demands.

Given polynomial-time reductions from the BP problem to both subproblems, the LEDR problem is also NP-hard.                     □

# C  Privacy Performance Analysis

In this section, we theoretically and systematically analyze the privacy performance of GEES against a typical inference method, i.e., Bayesian inference. Bayesian Inference Attacks (BIA) [19] is a widely-used method in privacy games to estimate the user's actual location. With the Bayes rule, the probability can be calculated as follows:

$$Q(\hat{l} \mid l^*) = \frac{\pi(\hat{l}) \cdot \mathcal{P}(l^* \mid \hat{l})}{Pr\{l^*\}} = \frac{\pi(\hat{l}) \cdot \mathcal{P}(l^* \mid \hat{l})}{\sum_{l \in \mathcal{L}} \pi(l) \cdot \mathcal{P}(l^* \mid l)} \tag{18}$$

THEOREM 1. *Let $\zeta$ be the neighboring protected location set (NPLS) of location $l$, then the upper bound of posterior probability by Bayesian adversaries can be obtained as $e^\epsilon \cdot \frac{\pi(l)}{\sum_{x \in \zeta} \pi(x)}$.*

PROOF.

$$\begin{aligned} Pr(l \mid l^*) &= \frac{\pi(l)\mathcal{P}(l^* \mid l)}{\sum_{x \in \mathcal{L}} \pi(x)\mathcal{P}(l^* \mid x)} \\ &= \frac{\pi(l)\mathcal{P}(l^* \mid l)}{\sum_{x \in \zeta} \pi(x)\mathcal{P}(l^* \mid x) + \sum_{x \in \mathcal{L}\setminus\zeta} \pi(x)\mathcal{P}(l^* \mid x)} \\ &\leq \frac{\pi(l)\mathcal{P}(l^* \mid l)}{\sum_{x \in \zeta} \pi(x)\mathcal{P}(l^* \mid x)} \\ &= \frac{\pi(l)}{\sum_{x \in \zeta} \pi(x)\mathcal{P}(l^* \mid x)/\mathcal{P}(l^* \mid l)} \end{aligned} \tag{19}$$

According to the definition of differential privacy in (1), and $0 < e^{-\epsilon} < 1$, we can also have that $e^{-\epsilon} \leq \frac{\mathcal{P}(l^* \mid l)}{\mathcal{P}(l^* \mid x)} \leq e^\epsilon$

Therefore, the above can be derived as:

$$\begin{aligned} Pr(l \mid l^*) &\leq \frac{\pi(l)}{\pi(x) + e^{-\epsilon} \cdot \sum_{x \in \zeta, \, x \neq l} \pi(x)} \\ &\leq e^\epsilon \cdot \frac{\pi(l)}{\sum_{x \in \zeta} \pi(x)} \end{aligned} \tag{20}$$

□

The upper bound of the posterior probability in (20) ensures users' absolute privacy by narrowing the effectiveness of inferences from the untrusted servers. This implies that regardless of prior knowledge, differential privacy can protect user privacy within the budget $e^\epsilon$. Therefore, it can guarantee the effectiveness and robustness of differential geo-obfuscation.

Next, we consider the lower bound of privacy gain by inference, representing the worst case in which the edge server surmises user privacy within NPLS.

THEOREM 2. *The lower bound of privacy $\mathcal{G}$ of a user is:*

$$\| \min_{\hat{l} \in \mathcal{L}, \, s_j \in \mathcal{J}(\hat{l})} \sum_{l \in \zeta} \frac{Pr(l \mid l^*)}{\sum_{x \in \zeta} Pr(x \mid l^*)} \cdot \frac{dist(\hat{l}, l)}{c_j} \|_2^2 \tag{21}$$

PROOF. Let $z$ be the optimal inferred location, represented by:

$$z = \arg\min_{l \in \mathcal{L}} \sum_{x \in \zeta} \frac{Pr(l \mid l^*)}{\sum_{x \in \zeta} Pr(x \mid l^*)} \cdot dist(\hat{l}, l) \quad (22)$$

Then, the inference error of (4) becomes

$$
\begin{aligned}
\mathcal{V}_k &= \sum_{x \in \zeta} \frac{Pr(l \mid l^*)}{\sum_{x \in \zeta} Pr(x \mid l^*)} \cdot dist(z, l) \\
&= \sum_{x \in \zeta} \frac{\pi(l) \cdot \mathcal{P}(l^* \mid l)}{\sum_{x \in \zeta} \pi(x) \cdot \mathcal{P}(l^* \mid x)} \cdot dist(z, l) \\
&\overset{(1)}{\geq} e^{-\epsilon} \cdot \sum_{l \in \zeta} \frac{\pi(l)}{\sum_{x \in \zeta} \pi(x)} \cdot dist(z, l) \\
&\geq e^{-\epsilon} \cdot \min_{\hat{l} \in \zeta} \sum_{l \in \zeta} \frac{\pi(l)}{\sum_{x \in \zeta} \pi(x)} \cdot dist(\hat{l}, l)
\end{aligned}
\quad (23)
$$

Therefore, the lower bound of privacy $\mathcal{G}$ of a user is derived as:

$$\mathcal{G} \overset{(5)}{\geq} \parallel e^{-\epsilon} \cdot \min_{\hat{l} \in \zeta,\ s_j \in \mathcal{J}(\hat{l})} \sum_{l \in \zeta} \frac{\pi(l)}{\sum_{x \in \zeta} \pi(x)} \cdot \frac{dist(\hat{l}, l)}{c_j} \parallel_2^2 \quad (24)$$

Theorems 1 and 2 collectively delineate the capability of differential geo-obfuscation method of GEES in countering Bayesian adversaries, which is contingent on the prior distribution of users in NPLS. This signifies that geo-obfuscation based on differential privacy can safeguard user privacy. Simultaneously, edge servers can enhance the overall service performance by gaining a certain degree of knowledge regarding users through Bayesian inferences with observations and prior knowledge. □

## D  Baselines in Differential Privacy

**Laplace.** Laplace-based differential obfuscation mechanism [24] introduces Laplacian noise to users' actual location. Mathematically, this will be expressed formally as:

$$P(l^* \mid l) \propto e^{-\epsilon \frac{dist(l, l^*)}{\mathcal{D}(\mathcal{L})}} \quad (25)$$

where $dist(l, l^*)$ represents the distance between the actual location $l$ and a nearby obfuscated location $l^*$, $\mathcal{D}(\mathcal{L})$ is the maximum distance between any two locations in the target area $\mathcal{L}$, and $\varepsilon$ is a privacy budget.

**Gaussian.** The Gaussian mechanism[27], as a relaxed mechanism in differential privacy, perturbs an actual location by adding noise sampled from a Gaussian distribution, which is mathematically represented by:

$$P(l^* \mid l) \propto e^{-\epsilon \cdot \frac{dist(l, l^*)^2}{2\sigma^2}} \quad (26)$$

where $dist(l, l^*)$ represents the distance between the actual location $l$ and a nearby obfuscated location $l^*$, and $\sigma$ is the standard deviation of the Gaussian noise.

