# OpenReview forum: "GEES: Enabling Location Privacy-Preserving Energy Saving in Multi-Access Edge Computing"
_ACM.org/TheWebConf/2024/Conference — TheWebConf24 Oral_

### Official Review · Reviewer_upZt · 2023-11-19

**Novelty:** 6
**Technical Quality:** 6

**Review:**

This paper presents GEES, a demand response system managing edge servers to hosting web services in edge computing with the incorporation of differential geo-obfuscation. Specifically, the authors claim that GEES can save significant energy consumption while protecting web users’ location information. They have done several groups of experiments and the results show that GEES can achieve satisfactory location privacy protection while outperforming the enhanced state-of-the-art approach in energy saving. In addition, the authors also conduct a set of experiments without privacy protection, showing the significant advantage of GEES in energy saving. More comments are detailed below:
Strengths:
1.	The topic is interesting and important. Energy consumption problem is always a significant research topic in cloud and edge computing. This paper considers this problem from a different perspective: user location privacy. To achieve the optimal performance in energy saving, accurate location information of users is needed. However, this violates users’ willingness to protect their data privacy. This introduces the trade-off problem which is investigated in this paper.
2.	The research problem and the solution adopted in this paper are well motivated. As I mentioned above, this problem exists in real world and needs to be solved. The authors use a simple but useful example to demonstrate the price of protecting user privacy and why GEES is needed. In addition, the authors use experiments to demonstrate the reason that the idea of differential privacy is used in this paper, rather than other approaches, e.g., homomorphic encryption.
3.	Generally, the paper is very well written and easy to follow. I really enjoy reading this paper. The background information, such as preliminaries and edge privacy game process, is sufficient for non-expert readers in this area.
4.	The authors have provided theoretical analysis of GEES and published the experiment-related documents, enhancing its technical contributions and feasibility in real-world implementation.

However, there are still several items that can be improved for the presentation quality:
1.	I would suggest the authors add “edge demand response” as one of the key words.
2.	More studies on location privacy related to this work should be discussed in Section 6.2.
3.	The bars in Figure 6 are a little small. It is better to increase its size, and the size of Figure 7 is not the same as the sizes of Figures 6(a)-(d).
5.	The reference styles are slightly inconsistent. Some of them contain abbreviations like INFOCOM, while others do not.

**Questions:**

1.	It is clear that DDP has multiple phases running on different devices. However, there are no such indicators in SGR. Where is this algorithm executed?
2.	Is it possible that some edge servers use location inference while others do not? Will these settings impact the performance of GEES and others?
3.	What kind of data are used in the experiments in Figure 1? This is needed to reproduce the results in Figure 1.

**Reviewer Confidence:**

4: The reviewer is certain that the evaluation is correct and very familiar with the relevant literature

**Scope:**

4: The work is relevant to the Web and to the track, and is of broad interest to the community

---

### Official Review · Reviewer_4Ru9 · 2023-11-22

**Novelty:** 5
**Technical Quality:** 5

**Review:**

This paper focuses on addressing privacy and energy-saving challenges in the field of Multi-Access Edge Computing (MEC). Many existing works that aim to address the demand response problem require obtaining accurate user location information. The goal of this paper is to identify methods that preserve privacy while maintaining system utility and energy efficiency. The paper redefines the problem as the Location Privacy-preserving Edge Demand Response (LEDR) problem. To optimize in these two objectives, a geo-obfuscated edge energy-saving (GEES) system is proposed. The system consists of two algorithms: Deflected Distribution Positioning (DDP) and Secure Greedy Response. DDP calculates personalized geo-obfuscation for each user by solving the Optimal Transport (OT) problem between the obfuscated distribution and the real distribution. Secure Greedy Response algorithm is designed to find the optimal allocation scheme for response in the MEC system.

Pros:
1. The article addresses practical issues in the field and background of the problem, focusing on the privacy concerns faced by the methods proposed in the academic community for energy reduction.
2. The problem formulation is detailed. The proposed optimization problems and corresponding algorithmic solutions are appealing.
3. The experiments are conducted meticulously, and the synthetic dataset and implementation work are commendable. The results are also compared against multiple state-of-the-art approaches.

Cons:
1. The novelty of the specific methods is not particularly prominent, as the essence of the DDP algorithm utilizes Particle Swarm Optimization (PSO).
2. The mathematical formulas in the Problem formulation and Proposed system sections are written with many errors and lack rigor, making it difficult for readers to understand.
3. The graphical representation in the results section could be improved, as some of the figures are ambiguous and can lead to misinterpretation.

**Questions:**

1. Novelty of the Article. I appreciate your pioneering effort in applying differential privacy (DP), particularly suitable DP, in the Multi-Access Edge Computing (MEC) domain. The definition of the Optimal Transport (OT) problem in the Deflected Distribution Positioning (DDP) algorithm, along with the utilization of Particle Swarm Optimization (PSO), is commendable. However, the distinctiveness of these contributions needs further clarification. While the integration of these elements does suprisingly well , could you please provide a more detailed explanation of the specific innovative aspects of your approach that set it apart from traditional methods?
2.Rigor of Mathematical Formulas. The mathematical presentation in your paper requires meticulous revision for clarity and precision. The current state of the formulas significantly hinders my comprehension! For instance, the ambiguity in the notation used, such as in Equation 4 with the subscript "k" in "$V_k = \sum_{\hat{l}_k \in \mathcal{L}} Q(\hat{l}_k | l^*_k) \cdot ||\hat{l}_k - l_k||^2$", raises confusion. Similarly, in Algorithm 2, the notation "$u_o$" seems inconsistent – should it be "$u_k$"? Additionally, the direct subtraction of q dimension from h dimension is perplexing. This lack of clarity obstructs the understanding of the relationship between computing resources and resource demands. There are many instances where the mathematical expressions are not rigorous. A thorough review and correction of these formulations are imperative for the article's clarity and accuracy.
3. Presentation of Results. The graphical representation of results, particularly in Figures 6 and 8, poses a risk of misinterpretation. In Figure 6, the separate labeling of color depth for system utility and energy efficiency complicates the comparison and interpretation. A more unified or clearer labeling approach could enhance comprehension. Regarding Figure 8, while detailed, the necessity to zoom in to discern each line suggests a need for a more reader-friendly presentation. Visual clarity and ease of interpretation are crucial for effectively communicating research findings. Could you consider revising these figures, possibly adopting more intuitive visualization techniques to facilitate a better understanding of the results?
In summary, while your work in the MEC domain using DP and OT within the DDP algorithm is promising, enhancing the clarity of your mathematical formulations and refining the presentation of your results are essential steps to improve the comprehensibility and impact of your article.

**Reviewer Confidence:**

3: The reviewer is confident but not certain that the evaluation is correct

**Scope:**

3: The work is somewhat relevant to the Web and to the track, and is of narrow interest to a sub-community

---

### Official Review · Reviewer_jyRH · 2023-11-22

**Novelty:** 5
**Technical Quality:** 6

**Review:**

This is very interesting work; I enjoyed reading this paper. The authors take a nice approach in that they are looking to balance user privacy with energy consumption on edge servers, which are both very important topics moving forward.

For a relatively short paper I found the experiments to be sufficient. The authors do a good job fitting analysis into the work and providing comparisons to several other techniques.

I found Figure 6 difficult to parse. It appears as though the main benefit that GEES introduces is privacy, while the other approaches are similar in terms of energy efficiency and system utility. I’d prefer to see the individual components of the bar graphs compared side-by-side. Also the legend spread across several figures is very confusing.

I found it surprising that a paper focused on adding privacy did not dive into attacks and defenses against the privacy they are adding. I’d like to see a detailed threat model and specific goals for the location privacy enhancements that they propose.

Pros:
* This paper is focused on a very real problem, and one that is increasingly relevant.
* I enjoyed that the authors look to find a balance between user privacy and the backend systems that users rely on, it isn’t one-sided.

Cons:
* Some of the presentations of results could be cleaned up for readability (e.g., Figure 6)
* I’m genuinely interested in getting the authors’ thoughts on location fuzzing in dense edge deployments (see question below). A part of me thinks that this may all be for naught given that edge servers will inherently have a decent idea of a user’s location naturally.

**Questions:**

I’m curious to know what the authors’ view is on the natural localization of edge servers in modern wireless networks. As they say in the intro, the density of edge servers in real-world 5G deployments can be extremely high. In that case, do the servers not inherently have a very good idea regarding users’ locations? What amount of location noise is substantially valuable in a privacy sense? I don’t believe that there is a concrete answer to this question, but I’m curious about the authors’ thoughts.

**Reviewer Confidence:**

2: The reviewer is willing to defend the evaluation, but it is likely that the reviewer did not understand parts of the paper

**Scope:**

3: The work is somewhat relevant to the Web and to the track, and is of narrow interest to a sub-community

---

### Official Review · Reviewer_5KUE · 2023-11-30

**Novelty:** 4
**Technical Quality:** 4

**Review:**

The paper addresses the energy consumption challenges associated with 24/7 running of edge servers in the context of the global deployment of the 5G network. It highlights the growing need for low service latency and the environmental concerns related to energy consumption and carbon emissions. The proposed solution, named GEES, focuses on achieving energy-efficient edge resource provision in the multi-access edge computing (MEC) architecture while addressing the privacy concerns associated with accurate location information of edge web users.

Strengths：
The paper is overall clear, well-written, and well-motivated.
The evaluation is clear and strong. The results show the potential and benefits of GEES compared to the other approaches.

Weakness：
Some aspects of the paper aren't clear (See the Questions section below)

**Questions:**

1. In Section 3.1, you mention the strategies of users and edge servers in the game, but there isn't sufficient detail in the paper to understand the essence of these strategies. To enhance comprehension and evaluation of the research contribution, it is recommended to provide a detailed description or a clear definition in the paper, explaining what specific strategies users and edge servers are choosing in the game.

2. In the Evaluation, emphasize why GEES outperforms other methods regarding system overhead. Offer a more in-depth discussion of factors that might influence the results, strengthening readers' understanding.

3. It is suggested that the authors consider validating GEES in terms of system utility and energy efficiency by constructing a prototype of a real system.

**Reviewer Confidence:**

3: The reviewer is confident but not certain that the evaluation is correct

**Scope:**

3: The work is somewhat relevant to the Web and to the track, and is of narrow interest to a sub-community

---

### Decision · Program_Chairs · 2024-01-22

**Decision:**

Accept (Oral)

**Comment:**

The paper introduces GEES, a solution addressing energy consumption challenges in running edge servers within the global deployment of 5G. Focusing on the MEC architecture, GEES aims for energy-efficient resource provision while addressing privacy concerns associated with accurate location information of edge web users (thus, the solution is NOT one-sided). The proposed solution employs geo-obfuscation algorithms, Deflected Distribution Positioning (DDP) and Secure Greedy Response, to preserve location privacy without compromising energy efficiency and system utility. Through thorough experiments, the paper demonstrates GEES's potential advantages in both energy savings and location privacy protection compared to existing approaches.

 The reviewers were all positive about the paper, and the feedback was consistent.
 They all acknowledged that the evaluation of the GEES solution is very well performed. Some suggested to build a prototype of a real system for validating the approach -- I find this comment extreme, and would absolutely not expect that the authors actually deliver this towards a camera ready version.
 The authors did reply to the reviews they received during the rebuttal phase, providing a way forward towards addressing the comments they received.

 I recommend accepting this paper, as it is tackling a very relevant and very challenging issue, which is of interest to the community.